# Prophylactic Peritoneal Fenestration during Kidney Transplantation Can Reduce the Type C Lymphocele Formation

**DOI:** 10.3390/jcm10235651

**Published:** 2021-11-30

**Authors:** Mohammad Golriz, Mohammadsadegh Sabagh, Golnaz Emami, Sara Mohammadi, Ali Ramouz, Elias Khajeh, Omid Ghamarnejad, Christian Morath, Markus Mieth, Yakup Kulu, Martin Zeier, Arianeb Mehrabi

**Affiliations:** 1Department of General, Visceral, and Transplantation Surgery, University of Heidelberg, 69120 Heidelberg, Germany; Mohammad.Golriz@med.uni-heidelberg.de (M.G.); Mohammadsadegh.Sabagh@med.uni-heidelberg.de (M.S.); Glnz.emami@gmail.com (G.E.); Sara.mohamadi.kh@gmail.com (S.M.); Ali.Ramouz@med.uni-heidelberg.de (A.R.); Elias.Khajeh@med.uni-heidelberg.de (E.K.); Omid.ghd@gmail.com (O.G.); Markus.Mieth@med.uni-heidelberg.de (M.M.); Yakup.Kulu@med.uni-heidelberg.de (Y.K.); 2Department of Nephrology, Heidelberg University Hospital, 69120 Heidelberg, Germany; Christian.Morath@med.uni-heidelberg.de (C.M.); Martin.Zeier@med.uni-heidelberg.de (M.Z.)

**Keywords:** consensus, lymphatic complications, preventive fenestration, surgical complication

## Abstract

Lymphocele is a common complication following kidney transplantation (KTx). We aimed to evaluate the preventive effect of peritoneal fenestration during KTx in reducing lymphocele. From January 2001, the data of all KTx were prospectively gathered in our digital data bank. From 2008, preventive peritoneal fenestration was performed as a routine procedure for all patients with KTx. Between 2001 and 2008, 579 KTx were performed without preventive peritoneal fenestration. To compare the results between with and without peritoneal fenestration, the same number of patients after 2008 (579 patients) was included in this study. The pre-, intra-, and postoperative data of the patients in these two groups were analyzed and compared, especially regarding the postoperative different types of lymphocele formation. The mean recipient age was 52.6 ± 13.8, and 33.7% of the patients were female. Type C lymphocele was significantly lower in the group with preventive fenestration (5.3% vs. 8.8%, *p* = 0.014 for 31/579 vs. 51/579). Peritoneal dialysis and implantation of the kidney in the left fossa were independently associated with a higher rate of type C lymphocele (OR 2.842, 95% CI 1.354–5.967, *p* = 0.006 and OR 3.614, 95% CI 1.215–10.747, *p* = 0.021, respectively). The results of this study showed that intraoperative preventive peritoneal fenestration could significantly reduce type C lymphocele.

## 1. Introduction

Lymphocele formation is one of the most common surgical complications following kidney transplantation (KTx) [1]. Although lymphoceles are usually asymptomatic, they may present with heterogeneous symptoms depending on their size and location [2]. Lymphoceles can even result in graft dysfunction by compressing the kidney or vasculature [3]. Because of the high frequency and organ-threatening risk of this complication, different preventive methods, such as precise ligation of donor and recipient lymphatic vessels [2], polymeric sealants/hemostatic biomaterials [3,4], povidone-iodine [5], and intraoperative fluorescent lymphography [6], have been proposed in the literature. However, lymphocele formation after KTx remains a challenging complication [7,8].

It has been shown that lymphocele formation was unlikely among the kidney grafts that were placed intraperitoneally because of the absorption of fluid through the peritoneum [1,9]. However, the transplanted kidney is preferably placed extraperitoneally in the iliac fossa. Therefore, creating a peritoneal window between the extraperitoneal and intraperitoneal cavity is the surgical procedure to manage lymphocele formation after KTx [10]. In addition, peritoneal fenestration during KTx as a preventive method has been shown to decrease the incidence of lymphoceles in one RCT and a few retrospective studies [9,11,12,13]. However, it is still unclear how effective this preventive fenestration is in reducing the different types of lymphocele based on the new severity grading consensus [14,15].

The aim of this study was to evaluate the role of prophylactic peritoneal fenestration in reducing different types of lymphoceles.

## 2. Materials and Methods

Since 2001, the digital data bank for kidney transplantation was established at the Department of General, Visceral, and Transplantation Surgery at the University Clinic of Heidelberg. All data from the kidney transplant patients were prospectively gathered in this data bank. Between January 2001 and December 2007 and after exclusion of recipients under 18 years of age, multi-organ transplantations, and intraperitoneal transplantations, a total of 579 KTx were performed without preventive fenestration. From January 2008, all KTx patients underwent prophylactic intraoperative peritoneal fenestration to prevent lymphatic complications. In order to compare the results of KTx with and without peritoneal fenestration, the same sample size of brain dead KTx after 2008 (579 KTx) with prophylactic peritoneal fenestration was considered in this study. Therefore, prospectively gathered clinical data of 1158 KTx patients (579 KTx without peritoneal fenestration and 579 KTx with peritoneal fenestration) were analyzed. The pre-, intra-, and postoperative data of the patients in these two groups were analyzed and compared, especially regarding the postoperative different types of lymphocele formation.

### 2.1. Preoperative Data Collection

Preoperative data included donor-related factors, such as brain dead/living donor, gender, age, body mass index (BMI), side of donor kidney, and recipient-associated factors, such as age, gender, BMI, indication for KTx, type and duration of dialysis before transplantation, comorbidity, previous abdominal operation, previous KTx, previous nephrectomy, and side of KTx.

### 2.2. Surgical Team and Surgical Procedure

All transplantations were performed by the attending surgeons according to the standard manual of our center. All procedures were performed via an extraperitoneal approach and placement of the kidney graft in the left or right iliac region. End-to-side anastomosis of the kidney vessels was performed either on one of the common, internal, or external iliac artery/vein, and an extravesical ureterocystostomy was established. Fenestration of the peritoneum was performed before closure of the abdomen in the first 579 patients after 2008. For fenestration, an approximately 2-cm round window was made in the peritoneum, medial, and close to the graft hilum. The peritoneal edges were not sutured, and the edges were not clipped. One easy flow drain was inserted through the peritoneal window into the abdomen, and another easy flow drain was placed parallel to the kidney in the retroperitoneal space. Both easy flow drains were extracted through the abdominal wall and fixed to the skin.

### 2.3. Posttransplant Patient Care

All patients received triple immunosuppressive therapy after surgery consisting of cyclosporine or tacrolimus, methylprednisolone, and mycophenolate mofetil. Immunosuppressive medication was administered and tapered according to a previously published routine protocol for postoperative care [16]. All patients received a single postoperative dose of pentamidine (in the form of a mouth wash) followed by oral trimethoprim-sulfamethoxazole as a Pneumocystis jirovecii prophylaxis, which was continued for 6 months after KTx. If necessary, Cytomegalovirus prophylaxis with Valganciclovir was also administered and maintained at a blood titer for 3 months.

KTx recipients were monitored postoperatively in a visceral transplantation intermediate care unit. Postoperative daily follow-up data, including clinical visits, imaging, and laboratory findings, were collected from a prospectively maintained database. Ultrasound examinations were performed at each visit, and fluid collections were recorded in detail. After discharge, patients underwent identical imaging and laboratory examinations at 3rd and 6th months postoperative.

### 2.4. Assessment and Management of Lymphocele

Lymphoceles diagnosed during the first 6 months of follow-up after KTx were included in this study. Lymphocele formation after KTx was defined based on the validated definition and severity grading system proposed by Mehrabi et al. [14,15]. Lymphoceles were divided into three severity groups, A, B, and C, based on the management plan [14]. According to the severity grading of lymphatic complications after KTx, grade A does not require treatment or is treated with aspiration and has a minor and/or non-invasive impact on clinical management; in grade B, non-surgical interventions are required; and grade C requires surgery, and patient management is significantly affected. Non-surgical interventions were defined as the implementation of percutaneous external drainage, with or without sclerotherapy. Surgical intervention was defined as laparoscopic drainage of the lymphocele into the peritoneal cavity. Recurrence was documented in cases of reappearance of a lymphocele after previous non-surgical or surgical intervention.

### 2.5. Statistical Analysis

IBM SPSS Statistics for Windows, Version 22.0, (IBM Corp. Released 2013. Armonk, NY, USA) was used for statistical analyses. Categorical data are presented as proportions and percentages, and continuous data are presented as means ± standard deviations. Categorical data were compared using the chi-square test or Fisher’s exact test. Continuous data were compared using the Student’s *t*-test or Mann–Whitney U test. Univariate and multivariate logistic regression analyses were performed to determine the independent preoperative predictive factors of wound infection and wound complication. Variables with a *p*-value < 0.2 from the univariate analysis were included in the multivariate regression analysis. Results of univariate and multivariate analyses are reported as odds ratio (OR) with 95% confidence interval (CI). A two-sided *p* value less than 0.05 was considered significant in all analyses.

## 3. Results

### 3.1. Study Population

A total of 1158 KTx (579 with and 579 without intraoperative peritoneal fenestration) met our inclusion criteria. The donor and recipient characteristics of the two groups are listed in Table 1. The comparison showed no significant difference between the groups, excluding the donor type (living or deceased), donor gender, and rate of peritoneal dialysis. The mean recipient age was 52.6 ± 13.8 years. A total of 33.7% of the patients were female. The most common indication for KTx was glomerulonephritis (42.0%), followed by congenital renal disease (27.4%), and the majority of patients had a history of preoperative dialysis (93.0%), with a mean duration of 63.7 (1–274 months). The most common comorbidity was hypertension (75.2%). Diabetes mellitus was observed in 14.6% of recipients. KTx was performed on the right side in 50.3% of patients and on the left side in 49.7% of patients.

### 3.2. Postoperative Lymphocele Formation

Lymphocele was diagnosed in 157 patients (13.6%). This included 83 out of 579 (14.3%) lymphoceles in the control group and 74 of 579 (12.7%) lymphoceles in the fenestration group. In addition, according to the applied severity grading system, the rate of type C lymphocele was significantly lower in the group with peritoneal fenestration than in the group without peritoneal fenestration (5.3% vs. 8.8%, *p* = 0.014 for 31/579 vs. 51/579). This difference could mirror itself in the rate of grade B lymphocele, which was higher in the group with peritoneal fenestration (3.7% vs. 1.3%, *p* = 0.007 for 22/579 vs. 8/579) as a sign of the effect of peritoneal fenestration in shifting the type C lymphocele to type B. There were no postoperative complications regarding peritoneal fenestration.

### 3.3. Predictive Factors of Grade C Lymphocele after KTx

To investigate factors associated with grade C lymphocele formation, univariate and multivariate regression analyses were performed (Table 2). Univariate analysis revealed that preventive fenestration was significantly associated with a lower rate of grade C lymphocele (*p* = 0.021). Receiving a graft from a brain-dead donor (*p* = 0.041), previous abdominal operation (*p* = 0.050), and implantation of the graft in the left abdominal fossa (*p* = 0.007) were factors associated with a higher rate of grade C lymphocele in the univariate analysis. Multivariate analysis revealed that implantation of the kidney in the left fossa was associated with an approximately three-fold increase in the occurrence of grade C lymphoceles (OR 2.842, 95% CI 1.354–5.967, *p* = 0.006). Moreover, peritoneal dialysis increased the risk of grade C lymphocele (OR 3.614, 95% CI, 1.215–10.747; *p* = 0.021).

## 4. Discussion

Lymphatic complications are common complications following kidney transplantation, which cause discomfort for patients and result in re-hospitalizations, re-operations, and even secondary graft loss [1,17]. These complications also increase healthcare costs. Therefore, the prevention and minimization of lymphocele formation is important. In the present study, the impact of preventive peritoneal fenestration on lymphocele formation after KTx was evaluated, and the non-dependent predictive factors for severe lymphocele (grade C) after KTx were studied.

Lymphocele after KTx is the most common peri-transplant fluid collection. Previous studies reported that up to 30–50% of postoperative perirenal fluid collections discovered by ultrasound examinations represent lymphoceles [18]. Fenestration of the peritoneum and internal drainage of lymph fluid into the peritoneal cavity are common treatment methods for symptomatic lymphoceles in many transplantation centers. The mechanisms of preventive fenestration are similar. Internal drainage of the lymph prevents accumulation of lymphatic fluid at the site of graft implantation [1,11,12]. However, few studies have investigated the outcomes of this prophylactic method [9,11,12]. A systematic review and meta-analysis by Mihaljevic et al. [12] assessed the effectiveness of this method and reported one non-randomized controlled clinical trial, one case series, and one randomized clinical trial. The analysis of the included studies showed that symptomatic lymphoceles decreased significantly in patients with preventive fenestration. Furthermore, no significant increase in postoperative surgical complications has been reported. However, some methodological flaws were mentioned in the study designs, including differences in endpoint definitions, relatively high selection bias, lack of standardization of the interventions, and inadequate follow-up length and protocols [12]. The above-mentioned RCT in 130 KTx patients showed that preventive fenestration is associated with a significant reduction in lymphocele after KTx from 15% to 3%. This study also showed that preventive fenestration reduces the necessity of invasive interventions following KTx [1,9].

Both recipient and donor vessels can be the cause of lymphatic leakage during vessel dissection, and no single surgical method or device has been utilized so far to prevent this complication. The surgical dissection and preparation method is different among the studies and centers. In our study, the combination of bipolar electrocoagulation, tie ligating with non-absorbable suture, and or clipping were applied. Electrothermal bipolar sealing device (LigaSure) has been used in some centers to decrease lymphatic leakage [19]. Other preventive procedures, such as the application of polymeric sealant or povidone iodine during the transplantation, have been described, but these were either not cost-effective, off-label, or did not decrease the incidence of lymphocele after KTx significantly [3,5]. Further studies with bigger sample sizes are necessary to evaluate the effectiveness of these methods.

Although few studies have discussed the role of peritoneal fenestration in reducing the rate of lymphocele formation after KTx [1,9,12,13], evaluating its impact with regard to the most recent grading consensus [14] has never been performed. In this study, for the first time, the role of intraoperative peritoneal fenestration in reducing the rate of postoperative lymphocele formation was discussed regarding the grading consensus. The results revealed that fenestration has a preventive effect on the incidence of grade C lymphocele. However, many surgeons are concerned about intestinal herniation or difficulties in performing biopsies following preventive peritoneal fenestration. Syversveen et al. [9] reported a non-significant increase in the rate of intestinal complications following peritoneal fenestration in their clinical trial [9]. In their trial, an incision with a length as long as the length of the transplanted graft was made in the peritoneum. In our study, we performed 2-cm peritoneal fenestration and experienced no postoperative intestinal herniation. On the other hand, smaller windows of the peritoneum increase the risk of early closure and subsequent lymphocele formation [20].

Regarding the prognostic factors, peritoneal dialysis and left-sided kidney transplantation were independent risk factors for postoperative lymphocele formation. Diplama et al. suggested that pre-transplant peritoneal stress, resulting from factors such as peritoneal dialysis, may increase the risk of post-KTx complications [21]. Traditionally, and due to technical preferences, such as a more superficial location of the external iliac vein, the right iliac fossa is preferred for kidney transplantation. Different technical factors, including previous abdominal operation and/or re-transplantation as well as the side of the donor’s organ, affect the surgeon’s decision on the implantation side [22]. It is still unclear if there is an absolute causal association between the side of KTx and lymphocele formation.

This study had some limitations. Beside the new developments in the field of transplantation, which make the comparison of old data challenging, one limitation is the retrospective design of the study. However, KTx patients were closely monitored and followed up in our clinic, and the details of each follow-up visit, examinations, imaging, and interventions were prospectively documented. Another limitation is the risk of selection bias in a non-randomized setting. Nonetheless, patients were assigned to the fenestration group according to standard preventive measures in our center, and the surgeons were not involved in patient selection, and all patients who underwent KTx were consecutively included in the study. As a result, it can be assumed that no patient-related factors influenced group allocations, and that bias was minimized.

## 5. Conclusions

In conclusion, preventive fenestration leads to significantly lower grade C lymphoceles without increasing the risk of other surgical complications, such as intestinal hernia. However, the findings of this study should be validated using randomized controlled trials.

## Figures and Tables

**Table 1 jcm-10-05651-t001:** Comparison of characteristics and post-transplant outcomes of the patients with lymphocele formation after KTx between preventive fenestration group and control group.

Donor	Preventive Fenestration N = 74	No Preventive Fenestration N = 83	*p*-Value
Brain dead/Living donor	55 (74.3%)/19 (25.7%)	73 (88.0%)/10 (12.0%)	**0.037**
Gender			**0.037**
Female/Male	44 (59.4%)/30 (40.6%)	35 (42.2%)/48 (57.8%)	
Age (years)	56.9 ± 12.0	53.1 ± 14.5	0.105
BMI (Kg/m^2^)	26.3 ± 5.0	25.4 ± 4.5	0.371
Side of donor kidney (Left/ Right)	34 (45.9%)/40 (54.1%)	38 (48.1%)/41 (51.9%)	0.867
**Recipients**			
Gender (Female/Male)	26 (35.1%)/48 (64.9%)	27 (32.5%)/56 (67.5%)	0.710
Age (year)	52.3 ± 14.6	52.8 ± 13.2	0.844
BMI (kg/m^2^)	25.5 ± 4.2	24.9 ± 3.9	0.384
Indication of KTx			0.985
Glomerulonephritis	32 (43.2%)	34 (41.0%)	
Congenital disease	19 (25.7%)	24 (28.9%)	
Diabetes/hypertension	8 (10.8%)	11 (13.3%)	
Obstructive nephropathy	2 (2.7%)	3 (3.6%)	
Vascular disease	2 (2.7%)	2 (2.4%)	
Tubulointerstitial renal disease	5 (6.8%)	3 (3.6%)	
Others/unknown	5 (6.8%)	6 (7.2%)	
**Comorbidities**			
Diabetes Mellitus	9 (12.2%)	14 (16.9%)	0.510
Hypertension	52 (70.3%)	66 (79.5%)	0.234
Heart disease	31 (41.9%)	33 (39.8%)	0.792
Previous abdominal operation	18 (24.3%)	24 (28.9%)	0.526
Previous KTx	7 (9.5%)	12 (14.5%)	0.388
Previous abdominal operation	17 (23.0%)	24 (28.9%)	0.490
Previous nephrectomy	10 (13.5%)	20 (24.1%)	0.154
Dialysis	70 (94.6%)	76 (91.6%)	0.442
Haemodialysis	59 (79.7%)	70 (84.3%)	0.471
Peritoneal dialysis	17 (22.9%)	9 (10.8%)	**0.040**
Preoperative dialysis (months)	57.1 ± 39.0 (1–140)	68 ± 49.8 (1–274)	0.163
Side of KTx (Left/Right)	39 (52.7%)/35 (47.3%)	40 (48.2%)/43 (51.8%)	0.549
Duration of operation (minutes)	195.3 ± 61.0	206.1 ± 56.0	0.304
Main immunosuppression			0.072
Ciclosporin/Tacrolimus	12 (16.2%)/62 (83.8%)	24 (28.9%)/59 (71.1%)	
**Lymphocele after KTx**			
Symptomatic	57 (77.0%)	63 (75.9%)	0.831
Recurrence	36 (48.6%)	37 (44.6%)	0.649
Graft rejection	1 (1.4%)	3 (3.6%)	0.457
**Grading of lymphocele after KTx**			**0.007**
Grade A	21 (28.2%)	24 (28.9%)	0.987
Grade B	22 (29.7%)	8 (9.6%)	**0.001**
Grade C	31 (42.1%)	51 (61.4%)	**0.014**
Duration of hospital stay	22.8 ± 15.3 (9–95)	27.6 ± 20.1 (6–165)	0.115

Abbreviations: BMI, body mass index; KTx, kidney transplantation.

**Table 2 jcm-10-05651-t002:** Univariate and multivariate logistic regression of predictive factors for grade C of lymphoceles after KTx.

Variables	Univariate			Multivariate		
	OR	95% CI	*p*	OR	95% CI	*p*
Donor age (≥60 y/o)	0.845	0.213–3.353	0.556			
Donor gender (male vs. female)	1.434	0.666–3.087	0.233			
Donor BMI (≥30 kg/m^2^)	1.179	0.540–2.578	0.413			
Brain dead/ Living donor	0.397	0.164–0.963	0.041	0.590	0.222–1569	0.290
Reciepient age (≥60 y/o)	1.079	0.496–2.350	0.500			
Reciepient gender (male vs. female)	0.823	0.373–1.819	0.387			
Reciepient BMI (≥30 kg/m^2^)	0.903	0.273–2.981	0.566			
Cardiovascular diseases	0.699	0.316–1.547	0.247			
Hypertension	1.366	0.537–3.471	0.340			
Diabetes mellitus	0.659	0.207–2.098	0.341			
Hepatic diseases	1.774	0.775–4.061	0.127	1.145	0.439–2.985	0.782
Previous abdominal operation	2.156	1.002–4.645	0.050	1.824	0.794–4.189	0.156
Previous KTx	0.885	0.271–2.887	0.552			
Hemodialysis	0.716	0.175–2.937	0.444			
Peritoneal dialysis	1.003	0.995–1.010	0.084	3.614	1.215–10.747	**0.021**
Nephrectomy	1.000	0.386–2.588	0.605			
Side of KTx (Left/ Right)	2.531	1.295–4.946	0.007	2.842	1.354–5.967	**0.006**
Preventive fenestration	0.453	0.232–0.886	0.021	0.334	0.153–0.730	**0.006**
Immunosuppression (Ciclosporin vs. Tacrolimus)	0.577	0.247–1.348	0.147	0.606	0.232–1.586	0.308

Abbreviations: OR, odds ratio; CI, confidence interval; KTx, kidney transplantation.

## Data Availability

By rational request from contributing author, the data can be provided.

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
