# Peer review of "Prophylactic Peritoneal Fenestration during Kidney Transplantation Can Reduce the Type C Lymphocele Formation"

_jcm, 2021, doi:10.3390/jcm10235651_

Round 1

Reviewer 1 Report

The article is well written and interesting. The author did a retrospective evaluation of a well known approach to prevent lyphocele after kt. 

I would suggest only to discuss about the role of the new technological devices for tissue dissection in preventing lymphocele.

Reviewer 2 Report

  1. the 2001-2007 cohort is old and there have been significant changes in transplant practice after that.
  2. what is PKTL grading? need to explain in method and also give abbreviation in the table.
  3. Table 2, the author should give all factors data in univariate and multivariate, not only the 5 factors
  4. In my opinion, the implications of the study are underdeveloped and must be improved and explained further in the final discussion.
  5. Words in the title are not usually in the keywords. In addition, the keywords are often written in alphabetical order.

  6. The authors must check the use of all acronyms, abbreviations, and notations employed in the whole manuscript.

  7. The manuscript needs to be proofread by the authors.

  8. The novelty and contribution of this study must be clearly stated.

Round 2

Reviewer 2 Report

Authors welcomed all suggestions and observations comprised in the first revision of the paper.